# Christian Shame and Religious Trauma

## Alison Downie

Department of Philosophy and Religious Studies, Indiana University of Pennsylvania, Indiana, PA 15705, USA; adownie@iup.edu

**Abstract:** The analysis of religious trauma is enriched by considering how it may be produced by formation in chronic shame. The testimony of those who have experienced religious trauma and severe religious shame is essential to interdisciplinary understanding of and response to this harm. The experiences of those harmed indicates that some traditional Christian doctrinal interpretations are shaming. Thus, the potential for Christian communities to create climates of chronic shame and cause religious trauma is present wherever such theological interpretations dominate. In this way, the religious teachings themselves, especially when communicated in chronically shaming environments, are traumatizing. In this approach, Christian religious trauma is not an added element to traumas of domestic, physical, or sexual abuse by a religious person or leader. Instead, the source of the trauma is formative experience of participating in Christianity. Religious trauma merits interdisciplinary study in Religious Studies and trauma studies, as well as Christian theology. Theological response to Christian religious trauma contributes to this interdisciplinary need.

**Keywords:** shame; Christian shame; religious trauma



## 1. Introduction

Scholarship investigating how religious teachings and practices may traumatize remains scant. Though the scholarship is sparse, religiously traumatizing experiences are not. Psychotherapist Dr. Alyson M. Stone has noted a striking disjunct between the silence of academic literature on the intertwining of religion and trauma and the religious harms evident in her clinical practice. According to Stone, "Religious trauma is more prevalent than the research suggests and often is a contributing factor to many of the problems that bring people to therapy, including depression, anxiety, and relationship difficulties. For this reason, religious trauma deserves careful attention" (Stone 2013, p. 324). Nearly a decade after Stone called for more study, the need remains. Religious trauma merits attention not only from psychologists and therapists but also from religious studies scholars, in both etic and emic approaches., This study employs an emic approach of constructive theology, in which resources inside and outside of the tradition are used to further analysis, develop critique of an identified problem, and suggest further development. After defining religious trauma, I will argue that theologian Stephanie Arel's analysis of chronic Christian shame elucidates one way in which religious trauma occurs in Christian contexts.

## 2. Defining Religious Trauma

Terms related to religious trauma, such as spiritual or religious abuse and spiritual violence, appear in related studies.[1] I use the term religious trauma for two reasons. Firstly, though definitions of and methodologies for studying religion are multiple and contested, in this context, the modifier religious is less definitionally fraught than the term spiritual, though contestation and overlap is inescapable. Spiritual harm may occur in any social context, from a workplace to a hospital to a community center. The modifier "spiritual" draws attention to an anthropological view of the human as inherently spiritual in some way and, therefore, potentially vulnerable to harm in that capacity by virtue of being human. In contrast, the term religious trauma is more circumscribed as a functional, descriptive

term. Religious trauma occurs within a religious context. Secondly, the term religious trauma clearly situates itself within interdisciplinary trauma studies. The term asserts that a specific form of trauma merits study as tied to a lived experience of a given religion, whether or not this experience also includes abuse or violence.

I will highlight three useful definitions of religious trauma, noting their strengths for later reference. Psychotherapist Marlene Winell (2012) coined the term "religious trauma syndrome" and maintains a website devoted to educating and helping people recover from it (https://journeyfree.org/; Accessed on 7 July 2022). According to Winell, religious trauma syndrome (RTS) is explicitly tied to "authoritarianism coupled with toxic theology" in Christian contexts, specifically in "doctrines of original sin and eternal damnation" (Religious Trauma Syndrome). I will build upon Winell's linkage of these specifically Christian teachings and religious trauma in later discussion of Christian shame.

Stone makes use of Winell's work but decouples religious trauma from a specific religious tradition, defining it as "pervasive psychological damage resulting from religious messages, beliefs, and experiences" (p. 324). In Stone's therapeutic experience, religious trauma often occurs in childhood and youth, during religious formation Stone argues that "unlike many forms of trauma that occur through acute incidents, religious trauma generally accrues gradually through long-term exposure to messages that undermine mental health" (p. 325). One of these messages is that emotions exist in a binary of good and bad, acceptable and unacceptable. In such contexts, bad emotions cannot be acknowledged or managed; they can only be condemned. According to Stone,

> "Prohibitions against entire categories of emotions can contribute to psychological difficulties, including depression, anxiety, guilt, and addictive or compulsive behaviors. In a similar fashion, the intellectual realm can become restricted, promoting legalistic, black-and-white thinking and difficulty with free association, fantasy, creative thought, and problem solving". (pp. 325–26)

Winell and Stone both argue that religious trauma occurs within a climate of rigid binaries, absolute judgments, and an atmosphere of fear of condemnation.

As trauma, the harm is not resolved simply by leaving a religious group or avoiding the religious tradition because trauma shapes bodies into a "fight/flight/freeze response" (Van der Kolk 2014, p. 54). As trauma researcher Bessel A. Van der Kolk (2014) puts it, "After trauma the world is experienced with a different nervous system" (p. 53). Stone argues that religious trauma produces symptoms of posttraumatic stress disorder (p. 326). These are generally recognized as including "intrusive memories, hyperarousal, hypervigilance, anxiety, depression, numbness, dissociation, compulsion to reenact, restriction of range of affect, and sleep disturbances" (Panchuk 2018, p. 509). In addition to PTSD symptoms, Stone also notes that those who have experienced religious trauma frequently find participating in religious services and events as intolerably distressful. At the same time, they often also employ "spiritual practices and beliefs to 'transcend' or deny problems rather than understand them", a mechanism called spiritual bypassing in some psychological literature (Stone 2013, p. 326). Religious trauma thus not only results in potentially lifelong PTSD symptoms, but also frequently shapes a way of being religious in which religious or spiritual teachings and disciplines are used to avoid rather than deepen self-awareness and healing.

Two essential contributions of Stone's work on religious trauma are the decoupling of its definition from a specific religious group or doctrine and her recognition that it may shape a relationship with religion which is opposite to the generally stated goals of spiritual and religious practices. Instead of religious or spiritual practices deepening and developing personal truthfulness, spiritual by-passing employs aspects of a religion to deny or avoid self-knowledge.

Finally, philosopher of religion Michelle Panchuk (2018, 2020) argues that the concept of hermeneutical injustice is significant for many experiences of religious trauma. Simply put, hermeneutical injustice occurs when oppressive social structures prevent knowledge and thereby diminish agency. Panchuk's argument is not limited to Christian contexts,

but she cites useful examples from this religious tradition. In one case, a woman raised in a Christian context is unable to recognize that she had been abused as a child (Panchuk 2020, p. 612). This is not because she does not understand the concept of child abuse in the abstract. Instead, it is because religious formation includes norms through which people interpret and understand their experience. A child who understands herself to be a sinner who deserves punishment is formed within a hermeneutic in which those who have authority to punish do so because it is deserved. Abuse is not an interpretive option. As an adult, the woman may still be unable to see abuse in her childhood experience, if her childhood religious hermeneutic framework remains intact. As Panchuk puts it, "certain theological commitments (for example, original sin or divine justice) can create an epistemic context in which particular instances of abuse cannot be recognized *as abuse* (220, p. 612). Panchuk (2020) also references a published account testifying to religious counsel a Christian church official gave to a woman that she ought to endure domestic abuse at the hands of her husband. The rationale of that counsel was that loving and following Jesus means to bear this suffering gladly.

Panchuk argues that both examples demonstrate the usefulness of the analytic lens of hermeneutical injustice for understanding religious trauma. In the first case, a woman is unable to know her own lived experience of abuse *as abuse* because of the religious context in which her knowledge was formed; in the second case, a religious authority taught a woman that she ought to accept domestic abuse for religious growth. According to Panchuk (2020), in both instances,

> "The victim is harmed as a knower because they are deprived of knowledge that they would otherwise have. They are harmed as an agent because this lack of knowledge prevents them from acting in accordance with their interests, goals, or values. Together, these two harms often result in other social and personal harms". (p. 613)

Panchuk's analysis is a significant contribution for understanding how religious trauma occurs. While the important psychological approaches of Winell and Stone focus upon trauma as individual, interior experience, Panchuk's study examines the larger context in which individual formation occurs, specifically the power of sanctioned knowing within a religious community. Panchuk's study emphasizes how religious trauma deprives persons of knowledge they might otherwise have and thus tends to emphasize harm through lackthrough not being provided what they need to form accurate assessments. Building upon these three explorations of religious trauma, I will argue that shamed persons' bodies are shaped to "know" what is false and, further, that this harm constitutes a form of religious trauma. To support this thesis, I turn to the study of chronic shame in Christian formation by Stephanie Arel (2016, 2019).

## 3. Shame

Shame, like trauma and like religion, is a multidisciplinary matter of study with vying definitional and methodological approaches. Arel employs an understanding of shame developed from within a relational anthropology of humans as bio/psycho/social beings. Following several affect theorists, Arel regards shame as one of several basic affects; at its core, it is a reflexive bodily response to a perceived social context. In this approach, some experiences of shame are inevitable, because the affect of shame is the flipside of the process of human bonding and forming attachments. Acceptance, a sense of belonging and secure connection result in effective bonding. But when an expected, anticipated experience of human connection is blocked, shame occurs. In this way, shame regulates social behavior through the discomfort of disconnection.

Developmentally, shame is experienced before language is established; some theorists suggest at about eighteen months (Arel 2019, p. 50). It is important to note that when shame occurs within a relational context of secure attachment, though always unpleasant, the pain of it may be transitory and not wounding in lasting ways. Within contexts of secure attachments, a child is socialized into norms of behavior by experiencing shame.

Occasionally painful rebuffs do not necessarily jeopardize a positive developing sense of self as valued and accepted if the child experiences these rebuffs within strong enough, positive enough relational context. As an occasional corrective, the discomfort of shame may facilitate connection in that the child learns what is not appropriate and can change behavior to avoid further rebuffs.

Once verbal ability develops, language entwines with bodily experience. The bodily sensations constituting the physiological response to blocked social connection—the sudden clenching of the gut, the red-hot, burning face-now have names, shame or humiliation. In language, the psychological experience entwines with the bodily experience.

In Arel's study, attending to the physiological dimension of shame is essential to understanding its formative power. As explained, in this approach, shame occurs in reflexive bodily processes before language develops; this means it functions in a child's body before reflective thought about the physiological responses are possible. Even after verbal ability develops, however, shame still occurs physiologically before thought in a specific moment is possible. According to this definition, shame happens physiologically and reflexively before a choice about response is possible. As capacity for reflective thought develops, perception of social context already formed in and by shame continues in shame's complex dynamic.

In contexts of insecure attachment and chronic shame, young bodies are shaped into a developing sense of self in deeply wounding patterns. To understand the power of this process of being formed in shame, Arel (2016) employs sociologist Pierre Bordieu's notion of *habitus* (p. 9). Shame can become more than an occasional response to an external stimulus. It can become a vicious cycle or *habitus*. In a *habitus* shame is not only externally imposed but also replicates and perpetuates itself in bodily response through perception, interpretation, and repeated patterns of relation (Arel 2016, p. 10). It becomes a bodily hermeneutic. For example, if a toddler is regularly mocked when expressing distress rather than comforted, the child's whole bio/psycho/social being, including bodily, reflexive responses, social expectation, perception, and developing sense of self will all be shaped in shame. In such a *habitus*, persons become ashamed of themselves and ashamed of being ashamed. Similarly, though theologian Stephen Pattison (2000) did not incorporate the notion of *habitus*, he also argued that it is very difficult for people to admit to feeling shame, even to themselves, and, therefore, also difficult to seek healing for it (p. 158). In this way, feeling ashamed of feeling shame blocks healing. In a vicious cycle, chronic shame fuels withdrawal and isolation as well as fearful, protective measures to avoid further exposure to shame, even exposure to self-knowledge of shame.

## 4. Chronic Shame and Religious Trauma

Using this understanding of what shame is and how chronic shame functions, Arel argues that, historically, Christian teaching has regularly advocated a religiously appropriate sense of self as one formed by shame. In large measure this is because Christian tradition has not distinguished between guilt and shame, even sometimes using the terms synonymously. Arel (2016) argues that Augustine's sense of self as "corrupt, immoral and depraved" reflects shame, not guilt (p. 73). A. Denise Starkey (2009) also argues that Augustine's *The Confessions*, particularly, is permeated by deep sense of shame (p. 101). Not only Augustine, but much of Christian theology, according to Arel, has explained human guilt and need for salvation in language which describes and elicits not guilt, but shame. Like Stephen Pattison (2000), Arel (2016) argues that shame has not received sufficient theological attention, with the result that theological understandings of the human person have not responded sufficiently to the reality of shame as "visceral, affective and neurological" (p. 11). Here is the nexus where shame and religious trauma converge as bodily wound.

According to Arel (2019), "Recent research on shame and trauma has more explicitly linked the two, locating shame at the core of symptomatology of Post-Traumatic Stress Disorder" (p. 55). Trauma-responsive theologies are a burgeoning area of investigation.[2]

Despite many resources in this material, however, these theologies have not addressed religious trauma and its relation to Christian formation in chronic shame.

As noted earlier, Winell's work emphasizes that in some Christian contexts, religious trauma is caused by religious doctrine. Testimonies of such childhood experience abound in many formats, though mostly outside scholarly literature. Panchuk (2020) brings some online sources into her work, citing a sermon which states "a baby is a 'viper in [a] diaper, a 'depraved', 'diseased' beast" (p. 617). The effect of being raised in this type of environment is evident in my own experience. I remember sermons about the indisputable proof of total depravity being children, who are bad in all natural inclinations and behaviors. I specifically remember the congregation's knowing chuckle in response to the minister's quip that "No one needs to teach a child how to lie". I also vividly recall a time of private prayer in which I tried with all my might to cry, to show God how sorry I was for being such a horrible sinner. I could not think of anything particularly bad I had done recently, but I knew without doubt I deserved hell. I was ten years old.

Tragically, formation in shame is not incompatible with sincerity of religious conviction or feelings of love by those teaching and shaming. In an interdisciplinary study of shame in Christian churches, Moon and Tobin (2018) state that "People often dispense this shame believing it will help their loved ones to conform to God's will and to spend eternity in heaven" (p. 453). Parents who feel love for their children may nevertheless, for many reasons, not create the bonding and secure attachment environment children need. Family and religious community dynamics may be structured by chronic shame, even when persons in such communities feel the emotion of love. Complicating matters further, systemic shame is often denied at individual, family, and community levels. The mechanism of spiritual by-passing Stone noted in the aftermath of an individual's religious trauma characterizes family and church ways of being religious as well. This dynamic is evident in generalizing statements that may be accurate aspirationally, but when asserted as description become shields of denial. For example, one Christian theologian writes that "Christians are always aware of the provisional nature of their own judgments about sin and righteousness" (Van Deusen Hunsinger 2001, p. 73). Such statements fail to distinguish between what Christian communities say they value and harmful behaviors that fall short of or even violate these values. The claim that Christians are already "always aware" by virtue of religious identity creates obstacles to becoming more self-aware by listening to those who may have been harmed within and by the community.

The most insidious aspect of Christian shaming is that it is routinely and consistently presented and justified as love. Blogger Samantha Fields (2021), referencing parental spanking throughout her childhood, writes, "I was taught that the people who hurt you, *violently* hurt you, every day, for years, are doing it because they *love* you". The significance of Panchuk's analytic of hermeneutical injustice in religious trauma becomes evident in Fields' statement. When children are taught that their experience of harm is not harm but an expression of love, hermeneutical injustice is clear. Those formed to believe that violent harm is an expression of love are thereby prevented from the possibility of considering their harm as abuse. They already "know" it is a demonstration of love. Shame can also be understood through a lens of hermeneutical injustice. In chronic Christian shame, those experiencing the pain of shame are also taught not to know what their bodies tell them, the truth of their own experience. This harm must be understood not only as deprivation of knowledge but as formation in a hermeneutical *habitus* of shame as a way of knowing encompassing all relation to self, others, world, and God.

Chronic shame shapes body-knowing. This sort of knowing is not that of concepts in the mind. It is a visceral knowledge, the knowledge of physical reflexes acting before the mind has time to suggest or consider alternatives. This knowing is a bio/psycho/social way of experiencing any and all aspects of life. Chronic shame shapes religiously traumatized body-selves to "know" they are utterly wrong, a wrongness which is not guilt or regret for a specific decision or behavior. Instead, it is an ontological wrongness of identity, the wrongness of self as self. The self shaped in and by shame thus knows itself not as subject

but as an object of condemnation (Pattison 2000, p. 72). Tobin and Moon (2019) develop a similar analysis in what they call "dispositional shame" in their ethnographic study of LGBT experiences in conservative Christian communities. They argue that "when shame becomes a person's habitual way of navigating the world, it violates the self and can harm . . . ability to cultivate virtues that support healthy relationships and moral action" (p. 113).

Theologically, formation in severe shame shapes a false knowing which prevents persons from knowing what Christian tradition generally purports to teach. Theologically, the religious trauma of chronic Christian shame is the harm of being prevented from knowing the truth of one's identity as being born beloved, loved into existence and sustained by Creator Spirit. Formation in shame prevents or obstructs persons from perceiving or listening to their own depths, to their own inner truth as relational subjects, not objects. Theologically, formation in chronic shame thus obstructs relationship to the Spirit by inculcating a way of being religious dependent upon fear and lack of genuine self-knowledge.

In its most severe outcomes, the only knowledge of self permitted in chronic Christian shame is that which conforms to the religious authority which establishes truth. All else is condemned as either sinful or false. In such contexts, truth is always and only external. Truth is received by submission to authority, by conformity. It is not known from within. Knowledge of self as subject in the sense of becoming aware of emotions, interests, desires, capacities, and aspirations that do not adhere to the religious norms is forbidden and obstructed not only externally but also internally through formation in a shame *habitus*.

Because it is such a painfully powerful affect, shame is an effective tool of control, a tool enforcing social conformity (Pattison 2000, p. 135). In a study of shame in the United States, Myra Mendible (2016) argues that "Historically the moral function of shame has tended to serve power rather than to challenge it" (p. 6). Shame functions this way in religious contexts as well as elsewhere. Those who would speak in religious settings about experiencing chronic Christian shame as religious trauma do not only risk encountering disagreement in the sense of a disputed thesis. Those speaking up within religious communities can expect the truth of their experience to be minimized or denied; they can also expect to be shamed back into silence in countless ways. Their character and religious commitments may be questioned, with implied or outright accusations that if their faith were stronger, their prayer more devout, or their understanding more spiritually mature, they would not have the experiences they describe. They may be accused of disloyalty, of betraying and harming family, community, or the larger religious tradition in sinful ways.

In an ethnographic study of a large Protestant group in the United States, Jessica Johnson (2018) documents multiple such shaming strategies. Johnson's analysis of one megachurch studies what she refers to, following theorist Sara Ahmed, as an "affective economy" in the church (p. 17). Johnson observes and analyzes the power of emotion at work in the community during services and events, one of which is shame. In one example, a leadership training program within the church was "conducted in the manner of bullying harassment" (p. 173). When one of the trainees expressed concern about the process, he was asked to have an interview in which church leaders told him his real problem was pride. He was grilled and told he had failed the interviewers. Johnson quotes the trainee's account at length, a portion of which states,

> "I started to feel like I was suffocating or like a giant weight was crushing me. I finally broke. The tears start running and at once my wife and I and everyone else in the room was relieved that I had admitted my sin. We were convinced that my concerns were irrelevant because I was prideful" (p. 174). Only after the session was the church member able to conclude that it had been "a manipulative experience". (p. 174)

Cashwell and Swindle (2018) refer to public shaming in religious contexts as "religious abuse", which they analyze as a form of "betrayal trauma" (p. 187). Their clinical perspective resonates with Panchuk's analysis of hermeneutical injustice when they state that those harmed may not "recognize the experience as abusive and traumatic" (p. 196).

Their clinical study also draws attention to the power dynamics at work in what they refer to as religious abuse and trauma (p. 189).

Precisely because shame is such a powerful tool of control, it is exceedingly difficult to bring it out into the open. As popular shame researcher Brené Brown (2012) puts it, "shame derives its power from being unspeakable" (p. 67). This makes theological naming and study of chronic Christian shame and religious trauma an urgent matter. Tobin and Moon (2019) study survivor accounts of clergy sexual abuse, one of whom is Ann Hagan Webb. Tobin and Moon write, "As horrific as the abuse was, Webb claims that the experience of being silenced, ignored, and even censured by members of the Catholic hierarchy when she came forward to testify to the abuse was equally if not more spiritually damaging" (p. 11). Panchuk (2018) has argued that "in order to pursue recovery, the survivor must be in a position to identify their experience for what it is—that is, they need access to the hermeneutical resources necessary to interpret their experience as *trauma*" (p. 523). The responsibility for naming such experience and changing power structures which not only harm but also protect those who harm others from accountability lies with the entire religious community, not with those who have been harmed. Theology has a place in this work to become a hermeneutical resource for new knowing, to counter formation in shame and religious trauma, to listen to witness and name the wounds, to speak against shame's fierce silencing.

Theologian Hilary Jerome Scarsella (2018) argues that theologies responding to trauma, particularly those by womanist, mujerista, and feminist writers, do not use the term recovery to discuss living in the aftermath of trauma. In this scholarship, what may be possible in trauma's aftermath is not recovery, but, instead, a new creation: "a process of making something new in the place of that which has been lost" (p. 269). For those formed in chronic shame, the process entails not recovery of a lost way of being self but forging a new way of being self-as-subject, rather than object, learning to discover what is within as worthy of being felt, acknowledged, considered, and perhaps followed. Theologically, this process entails openness to the movement and prompting of the Spirit arising within. Though it may involve concepts, this is not primarily an intellectual process but about felt experience in the body. As Marcia Mount Shoop (2018) writes,

> The truth of embodied trauma is that safety evaporates as a familiar sensation, but trust and openness do not have to disappear. They can be cultivated, they can be relearned, but the body must be the primal site of the learning. These are not simply ideas; they are visceral sensations that require cellular address. ("Body-wise", p. 246)

For bodies shaped in a hermeneutic of shame, recognizing harm as harm, not as expression of love, either human or divine, may be a most difficult but essential first step in this learning. Though this learning is not primarily intellectual, theologies which "reimagine the Sacred as a dynamic power, rather than an ageless being—a power that that [sic] dwells in earthly bodies (human and nonhuman) and makes itself known in the truths of our diverse, ever-evolving experiences" offer a promising frame for such reforming (Lelwica 2017, p. 211). The Christian claim that God is Love needs to be explored in the light of shame and trauma which prevent persons from experiencing love, from themselves, others, and God.

## 5. Conclusions

Theological response to chronic Christian shame and religious trauma is necessary for at least three reasons. Firstly, these wounds have not received the serious and sustained interdisciplinary and theological attention they merit. These wounds are not only individual, interpersonal matters, but also systemic, social, and structural issues requiring analysis from many quarters.

Secondly, integrating chronic Christian shame and religious trauma opens avenues for further discussion on intersections of similarity in shame-injury across difference. People are shamed within Christian contexts for a host of reasons beyond those mentioned here. In

addition, religious shaming may well intersect with broader cultural shaming of aspects of identity, as, for example Michelle Mary Lelwica (2017) argues in *Shameful Bodies*. Christian theological work testifying to and examining religious shaming may, in addition to breaking silence, open possibilities for interdisciplinary dialogue and deeper understanding of shame and trauma.

Finally, just as interdisciplinary study of trauma deepens understanding of how chronic shame and religious traumaharm, continuing empirical studies of bodily practices which help traumatized persons manage or reduce PTSD symptoms may also yield theologically significant insight. Christian theological understanding of the human must continually be shaped by and respond to developing interdisciplinary knowledge regarding chronic shame and religious trauma. For all these reasons, at least, theology has a place in and must engage with interdisciplinary response to chronic shame and religious trauma.

**Funding:** This research received no external funding.

**Data Availability Statement:** Not applicable.

**Acknowledgments:** I thank Kevin Timpe for making me aware of Michelle Panchuk's work and I thank Michelle Panchuk for making me aware of blogger Samantha Fields.

**Conflicts of Interest:** The author declares no conflict of interest.

## Notes

1    (Cashwell and Swindle 2018; Fernández 2022; Johnson and VanVonderen 2005; Pasquale 2015; Pisilä 2022; Tobin 2019; Tobin 2016).

2    (Arel and Shelly 2016; Baldwin 2018; Beste 2007; Hess 2009; Jones 2009; Rambo 2010; Rambo 2017; Shoop 2010).

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
