# Peer review of "Christian Shame and Religious Trauma"

_religions, doi:10.3390/rel13100925_

Round 1

Reviewer 1 Report

This is an interesting piece of work, with some thoughtful points made regarding shame process within the context of Christian faith. Your article makes a contribution but I feel it needs strengthening because at present it reads more like an opinion piece.

Below, I outline how I consider the manuscript could be improved:

Abstract: Please provide a simple definition of religious trauma. Please also outline the framework for your review, e.g., state explicitly what your work will cover and summarise the findings and implications succinctly. At present, from reading the abstract I am not sure what argument you are trying to convey. This should all be outlined here and the reader should be clearly signposted regarding what to expect. For example, aims, methods, discussion.

Introduction: Please introduce and define key terms early on. Please add more references; at present there are claims which are unsupported, e.g., closing sentence line 14 and 16 and throughout. These claims need supporting references. There are many other points, which require referencing.

You have some great discussion later on regarding key definitions and a justification for their use. I would, however, like to see some more vivid examples of religious abuse included for the unfamiliar reader. I note that you include one example regarding domestic abuse on pg.6 but examples could be given before this when key definitions are presented. 

Thank you for sharing your own experiences of shame in the context of your Christian upbringing, you make some important points and provide a helpful pathway to connect shame and the more damaging or extreme parts of Christian theology. In this section, however, I feel some of the theoretical claims need to be teased apart. For example, the Christian tradition is large and diverse and I feel that this is not taken account of within this work. Not all Christian theology is shaming, in fact, there are some very helpful aspects. Furthermore, not all Christian traditions, or theologies interpret theology in these negative ways. I feel that this needs to be acknowledged clearly.

As well as reflecting on your own experience, I feel it would be helpful to bring in some empirical literature, in particular, qualitative research which has examined more negative aspects of Christian theology on mental health. A paragraph or two of this form of literature would provide a helpful connection between your discussion of shame and earlier discussion of religious trauma and help to contextualise your discussion.

Again, this is quite a strong remark: "Those speaking up within religious communities can expect the truth of their experience to be minimized or denied... also expect to be shamed back into silence in countless ways."  Why so? Please provide evidence? Would this be the case for everyone? Some of the remarks here read as opinion pieces and need to be more appropriately balanced.

Reviewer 2 Report

In this paper, the author seeks to argue that a certain analysis of chronic Christian shame, due to Arel, can illuminate one form that religious trauma can take in Christian contexts. This is an interesting thesis that merits further discussion. This paper is a good example of such work. It is also timely; religious trauma is a burgeoning topic of research at the intersection of philosophy and religious studies. It should be of interest to researchers in analytic theology, religious studies, and trauma studies.

The argument is persuasively written – as a reader I found myself sympathetic to a lot of the key points in the argument. Its conclusion is supported by reference to a fairly broad array of interdisciplinary work, including trauma studies and religious studies.

At the level of prose, the paper was enjoyable to read. The writing was rigorous but without becoming overbearingly technical or jargon-filled. The author is to be commended for their thoughtful and respectful discussion of the sensitive subject matter of religious trauma.

The argument touches base with recent work in analytic theology, and this is best seen in the discussion of Panchuk’s work in Section 4. However, the argument may benefit from a slightly deeper discussion related work on religious trauma, such as Tobin’s paper ‘Religious Faith in the Unjust Meantime’ and Cockayne, Efird and Warman (2020) book chapter ‘Shattered Faith’.

Reviewer 3 Report

Review of  Christian Shame and Religious Trauma

This paper involves a perceptive account of how Christian formation can inculcate shame in individuals at a bodily, prereflective level. It is particularly good in showing how the hermeneutical framework Christianity can produce in the victims of trauma prevent them from dealing with the problem, and in the end the author suggests a role that theology can play in liberating people from the trauma religions can inflict.

p. 1: recommendations for therapy seem to precede in the order of treatment of shame that shows how the trauma occurs, shouldn’t it be the other way around?

Distinction between religion and spiritual, the latter presupposes an anthropology which asserts a spiritual dimension, wouldn’t the same be true of religion? Is the issue really the institutional and cultural dimension of religions, particularly Christianity in this paper?

Line 63 formations (why plural?)

The idea that trauma takes place over time, the formational life of a child, and the ways in which the author shows how religious teachings can be subsequently used by those traumatized to avoid healing are very good. The author shows how the very hermeneutical framework given by religions can block the idea that there even is such a thing as abuse at play.

Line 103, would it be good to discuss different religious traditions? Even Catholicism might have different versions, say between the Irish and Italian versions of it? This is just a question.

136, “harm through lack” could be developed a little, a few phrases

145, Shame is the flipside of bonding and attachment, please explain just a bit

163, isn’t it language than now becomes entwined with bodily experience?

171, reflexively here means something akin to a “bodily reflex” as opposed to “reflexive” in the sense of “reflective,” might be helpful to distinguish these at some place.

172, perception of a social context becomes part… but won’t such perception of the social context be there even before reflective thought develops?

183, “ashamed of being ashamed.” This is a very good point, a little development, a sentence or two might help explain it. A higher level of the same process of shame appears in regard to shame, and this leads to one being reluctant to talk with anyone about the trauma.

On p. 12, I wonder if similar mechanisms of shame are not often present in childhood formation in general. Infliction of religious trauma just builds on or gives a twist to other forms of shame-inducement often involved in the raising of children. The author alludes to such a point in line 289-290.

l. 251—in this paragraph, to deal with the good point on hermeneutic injustice in the first part of the paragraph, it would help to talk about how the traumatized person might use the fact that the trauma was inflicted in the name of love, from the traumatized person’s perspective. E.g. I should not even think that I was abused because whatever was done was done from love.

269. Good point, the true Christian tradition is about how we are loved as we are—this can help a person overcome earlier trauma (of all kinds, not just religious), it doesn’t justify the trauma though. The author wisely does not take a reductionistic approach to the religious tradition, seeing it as nothing other than abusive.

l. 326, why two periods

331, space after argued

360 dynamic power rather than ageless being, the two can go together and are not necessarily opposed, e.g. God saying I have loved you with an everlasting love.

379, separate trauma and harm 

Reviewer 4 Report

This is a fine essay exploring a topic which has not been give extensive consideration

Author Response

I thank Reviewer 4 for this positive evaluation

Round 2

Reviewer 1 Report

thank you for addressing the points raised and taking these on board. 

Author Response

I thank Reviewer 1 for the extensive specific comments I addressed in the first round. I believe those revisions make the paper much stronger. I acknowledge that Reviewer 1's concerns in this second round remain. I think I have addressed them as much as possible in the prior extensive revision and addition of citations.

Round 3

Reviewer 1 Report

Well done for taking the comments on board.